# Development of a Gene-Based Marker Set for Orange-Colored Watermelon Flesh with a High β-Carotene Content

**DOI:** 10.3390/ijms25010210

**Published:** 2023-12-22

**Authors:** Bingkui Jin, Gaeun Jang, Girim Park, Durre Shahwar, Jagyeong Shin, Gibeom Kwon, Yongjae Kim, Hoytaek Kim, Oakjin Lee, Younghoon Park

**Affiliations:** 1Department of Horticultural Bioscience, Pusan National University, Miryang 50463, Republic of Korea; 121971707@163.com (B.J.); silver2336@naver.com (G.J.); rlfla007@gmail.com (G.P.); dslr5400@gmail.com (D.S.); chzhqk29@naver.com (J.S.); 2Partner Seeds Co., Ltd., Gimje 54324, Republic of Korea; gibeom.kwon@partnerseeds.kr (G.K.); yongjae.kim@partnerseeds.kr (Y.K.); 3Department of Horticulture, Sunchon National University, Sunchon 57922, Republic of Korea; 4National Institute of Horticultural and Herbal Science, Rural Development Administration, Wanju 55365, Republic of Korea; ojlee6524@korea.kr; 5Life and Industry Convergence Research Institute, Pusan National University, Miryang 50463, Republic of Korea

**Keywords:** watermelon flesh, β-carotene, orange flesh, near-isogenic line

## Abstract

The fruit flesh of watermelons differs depending on the distinct carotenoid composition. Orange-colored flesh relates to the accumulation of β-carotene, which is beneficial to human health. Canary-yellow-fleshed OTO-DAH and orange-β-fleshed (orange-fleshed with high β-carotene) NB-DAH near-isogenic lines (NILs) were used to determine the genetic mechanism attributed to orange watermelon flesh. For genetic mapping, an F_2_ population was developed by crossing the two NILs. The segregation ratio of flesh color in the F_2_ population indicated that the orange-β flesh of the NB-DAH NIL was controlled by a single incompletely dominant gene. Through a comparative analysis of the whole-genome sequences of the parent lines and NILs, a major introgression region unique to the NB-DAH NIL was detected on Chr. 1; this was considered a candidate region for harboring genes that distinguish orange from canary-yellow and red flesh. Among the 13 genes involved in the carotenoid metabolic pathway in watermelons, only *ClPSY1* (*ClCG01G008470*), which encodes phytoene synthase 1, was located within the introgression region. The genotyping of F_2_ plants using a cleaved amplified polymorphic sequence marker developed from a non-synonymous SNP in *ClPSY1* revealed its relationship with orange-β flesh. The insights gained in this study can be applied to marker-assisted breeding for this desirable trait.

## 1. Introduction

Watermelon (*Citrullus lanatus* L.), cucumber (*Cucumis sativus* L.), pumpkin (*Cucurbita pepo* L.), and squash (*Cucurbita moschata Duchesne*) all belong to the Cucurbitaceae family. Originating in South Africa [1], watermelon is a fruit with a diploid chromosomal count of 2n = 22; its genome contains approximately 424 million base pairs (bp) and 22,571 genes [1]. Globally, watermelon is considered an important agricultural crop; for example, 1.1 million tons of watermelons were cultivated on 3.1 million hectares of land throughout the world in 2021 [2]. Despite watermelon’s global significance in agriculture, there remain specific challenges and gaps in our understanding of the fruit’s color. These include limited insights into the genetic mechanisms driving color variation, the environmental influences on pigment accumulation, and the interaction between genetic and environmental factors. Addressing these gaps is essential for breeding strategies aimed at improving fruit quality and consumer appeal.

Fruit color is an essential breeding trait that significantly affects consumer preference and the market value of the fruit in relation to visual appeal and the associated health benefits. Watermelon flesh ranges in color, including white, light yellow, canary-yellow, salmon yellow, orange, pink, red (scarlet or coral), and green [3,4], and the color is primarily attributed to the composition of carotenoids that accumulate in the chromoplasts [5]. For instance, the predominant red flesh of cultivated watermelons results from the accumulation of all-trans-lycopene, while the orange flesh primarily arises from the accumulation of either ζ-carotene and β-carotene or prolycopene (7,9,7’,9’-tetra-cis-lycopene) [3,4], and the canary-yellow flesh results from the synthesis of xanthophyll, including neoxanthin, violaxanthin, and zeaxanthin, from carotene via the carotenoid biosynthesis pathway [4,6,7]. The carotenoid composition in watermelon, contributing to the orange flesh color, is primarily due to the accumulation of ζ-carotene, β-carotene, and prolycopene. This accumulation is influenced by a combination of environmental factors, such as light exposure and temperature, and genetic factors, particularly the expression and functionality of enzymes like phytoene synthase (PSY) in the carotenoid biosynthesis pathway. Genetically, the preference for one biosynthetic pathway over another is governed by the expression and activity of key enzymes in the carotenoid pathway. Variations in genes encoding these enzymes, such as phytoene synthase, can shift the balance towards either ζ-carotene or β-carotene/prolycopene production, thereby influencing the fruit color and carotenoid profile of the watermelon flesh.

In the initial stages of carotenoid synthesis in the watermelon flesh, geranylgeranyl pyrophosphate (GGPP) within the plastids is catalyzed by the first enzyme involved in the carotenogenic pathway, phytoene synthase (PSY), to produce phytoene. DNA sequence variations in the phytoene synthase gene (PSY1), which influences the early stages of carotenoid synthesis, have been recognized as being the cause of color differences between orange-, red-, and canary-yellow-fleshed watermelon cultivars [8,9] by altering their activity or expression, affecting phytoene production and downstream carotenoid. Enhanced PSY1 activity can increase β-carotene or lycopene levels, intensifying flesh colors to orange or red, while reduced activity can lead to lighter colors like canary-yellow. Therefore, these variations, including point mutations, insertions, deletions, and regulatory changes, crucially dictate the watermelon’s flesh color by impacting the early stages of the carotenoid pathway. Subsequent to the action of PSY, a series of enzymatic reactions involving phytoene desaturase (PDS), ζ-carotene desaturase (ZDS), ζ-carotene isomerase (Z-ISO), and prolycopene isomerase (CRTISO) desaturate isomerize phytoenes to produce prolycopene and all-trans-lycopene. To date, loss-of-function mutations in CRTISO have been observed in *Arabidopsis* [10], tomatoes [11], and melons [12], leading to the accumulation of prolycopene. Similarly, the relationship between a non-synonymous single-nucleotide polymorphism (SNP) in CRTISO and the development of orange flesh with a high prolycopene content has been identified in watermelons [13]. In addition, the conversion of all-trans-lycopene into β-carotene is exclusively mediated by lycopene β-cyclase (*LCYB*). Mutations in the LCYB gene lead to the accumulation of all-trans-lycopene, which is associated with the formation of red flesh in watermelons [14]. Further downstream, β-carotene is enzymatically transformed into xanthophyll, which is primarily responsible for the occurrence of canary-yellow flesh. These conversions are facilitated by β-carotene hydroxylase (CHYB). However, although the presence of β-carotene in orange-fleshed watermelons is attributed to the function of CHYB, the relationship between the mutations in the CHYB gene and β-carotene accumulation has not been reported.

To determine the effect of a specific genetic mutation on a certain phenotype, individuals carrying the mutation can be compared with those that share an identical genetic background. However, identifying individuals with perfectly matched genetic profiles is challenging. In such situations, near-isogenic lines (NILs) offer highly efficient resources. NILs possess nearly indistinguishable genetic backgrounds, except for genes linked to the specific traits under investigation. These lines are typically developed by transferring the desired traits from a donor parent to a recipient parent through a series of repeated backcrossing steps. Subsequently, genome-wide comparisons of gene sequences between NILs and identification of introgressed chromosomal regions can be easily achieved through whole-genome resequencing (WGRS) using next-generation sequencing (NGS) technology. In the context of watermelon research, the comprehensive comparison of whole-genome sequences in NILs has been proven effective in identifying genomic regions associated with target traits, such as high trans-lycopene content, as demonstrated by Lee et al. [15].

Therefore, this study aimed to determine the genetic factors responsible for the manifestation of orange flesh with a high β-carotene content in watermelons (herein referred to as orange-β). Three distinct breeding lines were employed in this study: orange-β, canary-yellow, and red flesh, and the selection of the three distinct breeding lines in our study was decisive for exploring the genetic underpinnings of carotenoid accumulation and flesh color variation in watermelon. The orange-β line, rich in β-carotene, contrasted with the canary-yellow and red-flesh lines, provides a diverse genetic spectrum for analyzing carotenoid accumulation. By using a red-fleshed line as the recurrent parent, two NILs were meticulously developed to produce orange-β- and canary-yellow-fleshed lines. The primary investigative approach encompassed an in-depth comparative analysis of whole-genome sequences among the parent lines and NILs to identify introgression regions and potential candidate genes. To substantiate a genetic linkage between the candidate gene within the introgressed region and the orange-β trait, genetic mapping was diligently executed employing an F_2_ progeny.

## 2. Results

### 2.1. Analysis of β-Carotene Content

The β-carotene contents of the two NILs and their F_1_ progeny were determined via high-performance liquid chromatography (HPLC) analysis (Table 1). The average β-carotene content of the orange-fleshed NB-DAH NIL (BC_2_F_4_) was the highest at 135.4 mg/kg, while that of the canary-yellow-fleshed OTO-DAH NIL (BC_2_F_4_) was 15.5 mg/kg. The flesh color of the F_1_ progeny (NIL-F_1_) was almost similar to that of the OTO-DAH-NIL but slightly mixed with an orange color; its average β-carotene content was 44.2 mg/kg, which was slightly higher than that of the OTO-DAH NIL. The results indicate that the canary-yellow flesh and carotene content of the OTO-DAH NIL were superior to those of NB-DAH NIL.

### 2.2. Genetic Inheritance of Orange-β Flesh

The genetic inheritance mode for orange flesh with a high β-carotene content was determined by analyzing the segregation ratio of the flesh color in an F_2_ population derived from crossing the NB-DAH NIL and OTO-DAH NIL. Of the 89 F_2_ individuals (NIL-F_2_), 22 and 20 had canary-yellow flesh and orange flesh, respectively. The flesh color of the remaining 47 plants was a mix of canary yellow and orange (Figure 1); this phenotype was similar to that of the F_1_ plants, which exhibited a Mendelian segregation ratio of 1(22):2 (47):1(20) (χ^2^(0.05,1) = 0.30, *p* = 0.58). The results indicate that the orange flesh of the NB-DAH NIL is controlled by a single incompletely dominant gene.

### 2.3. Wole-Genome Resequencing (WGRS) and SNP Detection

WGRS of the parental lines and NILs was performed to detect genome-wide SNPs among the lines (National Center for Biotechnology Information (https://www.ncbi.nlm.nih.gov/) repository, accession number: SRR26208948, SRR26208949, SRR26208950, SRR26208951, SRR26208952). These SNPs were used to identify genomic regions in the NILs introgressed from the donors. The statistics of the raw and trimmed reads generated from the WGRS are presented in Table 2. The trimmed read values ranged from 20,030,848 (NB-DAH-NIL) to 48,997,839 (OTO-DAH-NIL), with an average of 31,448,467 reads. The average number of trimmed reads corresponded to approximately 65.82% of the average number of raw reads and provided coverage for approximately 15 folds of the Charleston gray watermelon reference genome (Table 2).

Genome-wide SNPs were detected by read-mapping the trimmed reads. The number of homozygous SNPs identified in each line compared with the reference genome ranged from 186,869 (NB-DAH NIL) to 219,027 (OTO-DAH NIL), with an average of 209,182 SNPs, thereby indicating similar values across the lines (Table 3). The chromosomal distribution of the identified homozygous SNPs was depicted based on a 1 Mb window for each line, and a similar pattern along the line was observed.

### 2.4. Detection of Introgression Regions in NILs

There were 147,343 homozygous polymorphic SNPs between the red-fleshed recipient DAH and orange-β-fleshed donor NB5410, and there were only 31,114 between DAH and orange-β-fleshed NB-DAH NIL. These results indicate that a large genomic amount of DAH was recovered in the NB-DAH NIL by backcrossing. A similar pattern was observed for the OTO-DAH NIL (Table 4).

There were 133,098 homozygous polymorphic SNPs between the donors NB5410 and OTO9491 and only 25,420 between the NILs; therefore, the NILs contained common genomes of their recurrent parent DAH, and the introgressed genomic regions associated with their flesh colors may differ (Table 4).

To identify the introgression regions from the donor parents in each NIL, the same SNPs as the donor parents in the NILs were selected from the polymorphic SNPs between the parents. The distribution of the SNPs on each chromosome was examined (Figure 2). Only homozygous SNPs with a depth > 3 were used for this analysis. Between the DAH and NB-DAH NIL, 31,114 polymorphic SNPs (Table 4 and Appendix A) were examined, and these were found to be clustered in chromosomes (Chr.) 1, 4, and 10 at high frequencies in the NB-DAH NIL (Figure 2, second track). In contrast, 16,755 polymorphic SNPs between DAH and OTO-DAH NIL (Table 4 and Appendix A) were examined. Polymorphic SNPs in the OTO-DAH NIL were located in regions of Chr. 4 at a high frequency (Figure 2, third tract); these regions were considered major introgression regions associated with fruit color, and further analyses were conducted based on these findings.

### 2.5. Detection of Introgression Regions Shared between the NILs

To investigate the donor-parent-derived introgression regions shared between the two NILs, the distribution of 9520 comparable SNPs within the introgression regions was examined (Appendix A). A total of 8481 SNPs were clustered on Chr. 4 (Figure 2, first tract), and a subsequent comparison of nucleotide sequences between the NILs revealed that all these SNPs were non-polymorphic. This finding indicates the presence of gene(s) within the introgression region on Chr. 4 that distinguish canary-yellow and orange-β flesh from red flesh.

### 2.6. Detection of Introgression Regions Distinct in Each NIL

Assuming that the introgression region(s) distinct in each NIL may contribute to differentiating between canary-yellow and orange-β flesh, polymorphic SNPs were determined that were specific to each NIL within their respective introgression regions. We identified 20,500 (Appendix A) and 3503 (Appendix A) polymorphic SNPs in the NB-DAH NIL and OTO-DAH NIL, respectively. The distribution of these SNPs across different chromosomes was examined, revealing major introgression regions in Chr. 1, Chr. 4, and Chr. 10 specific to the NB-DAH NIL (Figure 2). In the introgression regions specific to NB-DAH NIL, we observed 9853 SNPs in Chr. 1, 4556 SNPs in Chr. 4, and 4324 SNPs in Chr. 10 (Appendix A). These regions are considered candidate regions harboring genes that distinguish orange-β flesh from canary-yellow flesh.

### 2.7. Exploration of Carotenoid Biosynthesis Pathway Genes within the Introgression Regions

Thirteen genes involved in the carotenoid biosynthesis pathway were selected, and their physical positions were determined based on Charleston gray reference genome information (Table 5). We examined the presence of these genes within the introgression regions of each NIL. Our findings confirm that *ClPSY1* (*ClCG01G008470*) which encodes PSY1 (the enzyme that converts geranyl pyrophosphate (GPP) to phytoene) is located within the introgression region of Chr. 1 in the NB-DAH NIL (Table 5, Figure 2). Additionally, *ClLCYB* (*ClCG04G004090*) which encodes LCYB (the enzyme responsible for the conversion of all-*trans*-lycopene to β-carotene) was detected in the common introgression region of Chr. 4 in both NILs. However, the remaining genes in the pathway were not present in the introgression regions. This suggests that the *ClPSY1* gene on Chr. 1 and *ClLCYB* on Chr. 10 may be involved in the expression of orange-β flesh, while the *ClLCYB* gene on Chr. 4 is essential for the expression of both orange-β and canary-yellow flesh.

Based on the WGRS results, we analyzed the sequence variations in these three genes. For the *ClPSY1*, we identified a non-synonymous SNP of the transition from A to G in the first exon at 10,040,402 bp on Chr. 1, which converts lysine (Lys, AAG) into glutamic acid (Glu, GAG). In the case of the *ClLCYB*, three SNPs were found in the first exon at 15,694,806, 15,695,470, and 15,696,099 bp positions on Chr. 4. Among these, the SNPs at 15,694,806 bp were synonymous, while the SNPs at 15,695,470 bp and 15,696,099 bp were non-synonymous and changed valine (Val, GTC) to phenylalanine (Phe, TTC) and lysine (Lys, AAG) to asparagine (Asn, AAC), respectively. For the *ClLCYB* gene on Chr. 10, no SNP was identified in the cis-element regulatory and genic region (exon and intron), indicating that this gene and introgression region is unlikely associated with the expression of orange-β flesh. The non-synonymous SNPs located in the exons of *ClPSY1* and *ClLCYB* are summarized in Table 6.

### 2.8. Development of ClPSY1-Gene-Based CAPS Marker and Linkage Analysis

To investigate the genetic linkage between the *ClPSY1* gene and orange-β flesh, a CAPS marker was developed from the SNP (A>G) in *PSY1* (Table 7) and used for genotyping an F_2_ population derived from crossing NB-DAH NIL and OTO-DAH NIL. Among the 84 F_2_ plants, 22 plants with clear orange flesh and 20 with clear canary-yellow flesh were selected for genotyping, while the remaining 42 plants showing intermediate flesh color, such as the F_1_ progeny, were excluded from the genotyping because of ambiguity in precise phenotyping. The PSY1-based CAPS marker PSY1-A>G was designed to produce the PCR bands of 43 and 338 bp from the canary-yellow- and red-fleshed lines and the PCR bands of 43, 163, and 175 bp in the orange-β-fleshed lines when the PCR amplicons were restricted with *BcoD*I (Table 7). Genotyping of the NILs using the PSY1-A/G marker confirmed the expected banding pattern and validated the SNP information obtained through NGS (Figure 3). Furthermore, genotyping the F_2_ individuals revealed co-segregation between the marker genotype and flesh color; 20 orange-fleshed F_2_ plants showed the marker genotype for the NB-DAH NIL, while all 20 canary-yellow-fleshed F_2_ plants exhibited the marker genotype for the OTO-DAH NIL (Figure 3). Two F_2_ plants showed the marker genotype for heterozygosity. These results indicate that *ClPSY1* is most likely the gene associated with the expression of orange flesh with a high *β*-carotene content.

To validate the accuracy of the NGS results, one CAPS and one dCAPS marker were developed for two non-synonymous SNPs in the ClLCYB gene (Table 7) and analyzed for marker genotypes in five lines. A CAPS marker LCYB-G>T for the SNP at 15,695,470 bp was designed to produce 1180-bp and 414-bp bands in a red-fleshed line and a 1594-bp band in canary-yellow- and orange-fleshed lines when the PCR products were digested by the *BsaH*I restriction enzyme. The dCAPS marker LCYB-G>C designed for the SNP at 15,696,099 bp produced a 210-bp band in the canary-yellow- and orange-fleshed lines and a 182-bp band in the red-fleshed line when digested by the *Afl*II restriction enzyme. These CAPS and dCAPS markers were genotyped on five lines, and the results showed the banding patterns as expected and validated the SNP information by the NGS (Figure 4). Therefore, the *ClLCYB* gene on Chr. 4 is required for the expression of both canary-yellow and orange-β flesh.

### 2.9. Confirmation of CAPS Markers Using Diverse Breeding Lines

A total of 22 watermelon cultivars (including red-, canary-yellow-, orange-β-, and orange-p- fleshed cultivars) were genotyped using PSY1-A>G and LCYB-G>T markers to validate their usability for the marker-assisted selection of orange-β flesh. In addition, the *CRTISO* gene-based marker CAPS-C^1976^ [13] was evaluated for the selection of orange-p flesh with a high prolycopene content. The orange-β flesh of “Summer Orange” and orange-p flesh of “Golden Honey”, “Orange Flesh Tender Sweet”, and “Tender Gold” were reported by Jin et al. [13].

The results showed that all orange-fleshed cultivars (except for three orange-p-fleshed cultivars) exhibited the same genotype as NB5410 and NB-NIL for all three markers, whereas the remaining five red-fleshed and four canary-yellow-fleshed cultivars showed the same genotype as DAH and OTO9491 and OTO-NIL, respectively (Table 6, Figure 4).

In contrast, the three cultivars with orange-p flesh showed the genotype for red flesh in the PSY1-A>G and LCYB-G>T markers and the genotype for orange-p flesh in CRTISO-T/C. Therefore, red (AA, GG, TT), canary-yellow (AA, TT, TT), orange-β (GG, TT, TT), and orange-p (AA, GG, CC) flesh can be discriminated using these three markers (PSY1-A>G, LCYB-G>T, CRTISO-T>C), and four orange-fleshed cultivars with unknown carotenoid content may be high in β-carotene (Table 8).

## 3. Discussion

Carotenoids are active biological pigments that affect the economic value of watermelon fruit. The various flesh colors of watermelon result from the distinct composition of carotenoids that accumulate in the chromoplasts. It is acknowledged that the manifestation of orange-colored flesh may be due to the accumulation of β-carotene, which is a precursor to vitamin A and is known for its antioxidant properties and immune-support functions. Therefore, gaining insights into the genetic mechanisms governing the development of orange flesh with a high β-carotene content is important for enhancing the quality and nutritional benefits of watermelons.

In this study, we investigated two NILs characterized by distinct orange-β and canary-yellow flesh colors as well as an F_2_ population derived from these NILs. Based on the results of our comparative analysis of the introgression regions in the NILs and genetic linkage analysis, we found that *ClPSY1* on Chr. 1 was the most likely candidate gene responsible for the development of orange-β flesh. An SNP that alters lysine to glutamine at the 149th position in PSY1 was distinguished between the *ClPSY1* alleles that dictate orange-β and non-orange-β fruit flesh, and it was determined that the presence of the *ClLCYB* allele that dictates canary-yellow flesh is required for the expression of orange-β flesh. This inference was drawn from the enzymatic role of LCYB in converting all-trans-lycopene (responsible for the formation of red flesh) into β-carotene, which forms orange-β flesh through the carotenoid metabolic pathway. We also conducted genotyping on 22 different watermelon cultivars using the CAPS markers developed from non-synonymous SNPs detected in these two genes and CRTISO for orange-p flesh. Our results confirmed the potential application of these three gene-based markers in MAS for determining the carotenoid composition and flesh color of watermelons.

PSY plays a major rate-limiting role in carotenoid biosynthesis [16]. At the upstream of carotenoid biosynthesis, PSY condenses two GGPPs to produce phytoene, which is the first carotenoid product. Although the correlation between PSY protein levels and the carotenoid content has been reported in *Arabidopsis* and carrots [17], the mechanism by which *ClPSY1* regulates β-carotene accumulation in orange watermelon flesh is not fully understood. However, the mutant allele of *CRTISO*, which catalyzes the conversion of phytoene into red-colored lycopene, is associated with a high prolycopene content in a range of fruit flesh types. Similarly, the mutation in *CHYB*, which converts β-carotene into xanthophyll, may be responsible for β-carotene accumulation in watermelon flesh. However, no studies to date have supported such an inference. By conducting quantitative trait locus (QTL) mapping for orange-β flesh based on the genotyping-by-sequencing (GBS) results of F_2:3_, Branham et al. [18] identified a significant SNP at 8,973,472 bp on Chr. 1. This locus is located approximately 1.5 Mb upstream of *ClPSY* and 2.0 Mb downstream of *CHYB*. Although the results suggested *CHYB* as a candidate gene responsible for high β-carotene accumulation, the cloning and sequencing of this gene revealed no sequence variation in *CHYB* between the parents of the F_2:3_ population. Furthermore, in our results, *CHYB* was not located in the introgression regions of NB-DAH NIL, and no SNPs were detected in this gene among the parents; thus, *CHYB* may not be associated with β-carotene accumulation in orange watermelon flesh.

Recently, two independent studies suggested that *PSY1* is a candidate gene for orange flesh. Song et al. [19] conducted a comparative analysis of the whole-genome sequence of 24 watermelon inbred lines (six orange-, nine red-, and nine yellow-fleshed) and reported that the non-synonymous SNP (A>G) in the first exon of *ClPSY1*, which is the same SNP discovered in our study, exhibited an 87.5% matching rate to flesh color; the SNP (G) for orange color was present in all six orange-fleshed lines, two red-, and one yellow-fleshed lines. In addition, by performing a high-resolution genetic mapping of an F_2_ population derived from red- and orange-β-fleshed inbreds, Nie et al. [4] identified two carotenoid biosynthesis genes in the QTL intervals: *ClPSY1* on Chr. 1 and *RCCR* encoding red chlorophyll catabolite reductase (RCCR) on Chr. 2. The non-synonymous SNP (A>G) in *ClPSY1* was also detected by Nie et al. [4]. In higher plants, RCCR is essential for chlorophyll degradation during leaf senescence and fruit ripening. Although Nie et al. [4] suggested the possible role of *RCCR* in the regulation of lycopene and β-carotene biosynthesis, we were unable to determine a sequence variant in the coding region of this gene from the WGRS data of parental lines in our study.

In melons, the *CmOr* gene (*MELO3C005449*) encodes the orange protein that controls the accumulation of β-carotene, the predominant carotenoid in orange-fleshed melon fruits [20,21]. *CmOr* induces chromoplast formation with arrested β-carotene turnover in orange-fleshed melon fruits through the posttranscriptional regulation of *PSY*. An SNP that replaces arginine (Arg) with histidine (His) in the orange protein leads to a higher accumulation of β-carotene. Thus, a CAPS marker based on this SNP allows for differentiation between orange and green melon fruits. An orange gene-homologue (Orange-Orange, *Cla97C01G005050*) in watermelon is located at approximately 5.3 Mb upstream of *PSY1* on Chr. 1. However, this gene was not included in the introgression regions of NB-DAH NIL, and no SNPs were detected in this gene among the parents.

## 4. Materials and Methods

### 4.1. Development of Near Isogenic Lines (NILs) and F_2_ Population

The canary-yellow-fleshed (OTO-DAH NIL) and orange-β-fleshed (NB-DAH NIL) NILs were developed using inbred canary-yellow-fleshed OTO9491 and orange-β-fleshed NB5410 as the donor parents and red-fleshed inbred DAH as the recurrent parent (Figure 5). To develop each NIL, F_1_ plants obtained by crossing the donor and recurrent parents were backcrossed to the recurrent parent. Subsequently, a canary-yellow-fleshed or an orange-β-fleshed individual was selected from the BC_1_F_1_ generation, and the selected individuals were then backcrossed to the recurrent parent to produce the BC_2_F_1_ generation. Canary-yellow- and orange-fleshed BC_2_F_1_ individuals were selected for self-pollination. The selection of individuals and self-pollination were repeated for additional two generations to form the BC_2_F_4_ generation and develop genetically fixed NILs: canary-yellow-fleshed OTO-DAH NIL and orange-β-fleshed NB-DAH NIL (Figure 5). Thus, the OTO-DAH and NB-DAH NILs shared the genetic background of DAH while exhibiting canary-yellow and orange flesh, respectively.

For genetic mapping, an F_2_ population was developed by self-pollinating an F_1_ individual derived from crossing the OTO-DAH NIL and NB-DAH NIL (Figure 5). The NILs and the F_2_ population were obtained through artificial pollination in a greenhouse located at the breeding field of Partner Seed Company (Ansung, Republic of Korea) from 2015 to 2020.

### 4.2. Analysis of Flesh Color and β-Carotene

The seedlings of watermelon plants were grafted onto squash rootstocks (“Shin-toza”) at the first true-leaf stage. Twenty days after grafting, three to six plants per line were transplanted in a vinyl greenhouse (Changwon, Republic of Korea). The watermelons were cultivated using the typical summer cultivation method practiced in the Republic of Korea, where each plant is grown on the ground and produces one fruit. For flesh color and carotenoid analysis, the fruits were harvested 40 days after flowering, cut in half, and photographed, and four cubes (approximately 5 g of fresh weight) were sliced from both halves, starting from the center. After removing the seeds, flesh samples were freeze-dried using a vacuum freeze-drying system (Biocryos, Daegu, Republic of Korea).

For high-performance liquid chromatography (HPLC), the freeze-dried flesh samples were powdered, and 1.0 g of the powdered samples was used for each extraction. Three crushed samples from each variety were placed in a 50-mL Falcon tube, and 20 mL of acetone was added. The samples were then sonicated for 3 min to achieve complete homogenization. Following the complete homogenization of the samples, they were left undisturbed at a low temperature (4 °C) in the dark for 12 h. The supernatant was collected by centrifugation at 1500× *g* for 5 min, and the supernatant was dried using a rotary evaporator at 25 °C. Subsequently, the concentrated extract was dissolved in a mixture of methyl tert-butyl ether and methanol (1:1, 2 mL *v*/*v*) and passed through a 0.45 μm filter for HPLC analysis.

HPLC was performed using an Agilent Technologies 1100 Series HPLC system (Palo Alto, CA, USA) equipped with a degasser, pump, autosampler, and column oven. Separation was conducted on an MGII C18 column (4.6 mm × 250 mm, 5 μm; Shiseido, Tokyo, Japan) maintained at 25 °C. The mobile phase consisted of 90% acetonitrile (A) and 100% ethyl acetate (B). The linear gradient conditions were as follows: 0–16 min, 0–57% (B); 16–30 min, 57% (B); and 30–40 min, 57–100% (B). The flow rate was set at 1.0 mL/min, and the injection volume was 20 μL. β-carotene was identified and quantified based on the absorbance value at 450–471 nm. The standard for β-carotene was obtained from Sigma-Aldrich (St. Louis, MO, USA).

### 4.3. Analysis of Genetic Inheritance of Orange-β Flesh

The F_2_ plants were categorized based on their flesh color: “A” for NB-DAH NIL, “B” for OTO-DAH NIL, and “H” for the intermediate of these lines, similar to the F_1_ progeny. The expected and observed values were calculated in accordance with the Mendelian segregation ratios for a single dominant gene. To assess the fit of the genetic inheritance model, a chi-square test was performed using the CHISQ.DIST.RT function in Microsoft Excel v.1618 (Microsoft, Albuquerque, NM, USA).

### 4.4. DNA Extraction and Whole-Genome Resequencing (WGRS)

For WGRS, genomic DNA was extracted from young leaves collected at the second or third true-leaf stages. Leaf samples were rapidly frozen in liquid nitrogen and pulverized using a mortar and pestle, and DNA was subsequently isolated using the GenEx™ Plant kit (Geneall, Seoul, Republic of Korea) following the manufacturer’s instructions. The extracted DNA was stored at −75 °C until further use in subsequent analyses.

Sequencing libraries were constructed using 250 ng of genomic DNA per sample, and quality control analysis of the constructed libraries was performed using gel electrophoresis. The prepared library was purified using a Qiagen MinElute column (Hilden, Germany), and sequencing was performed using the Hiseq2000 NGS platform (Illumina, San Diego, CA, USA) in paired-end sequencing mode by Seeders (Daejeon, Republic of Korea). Sequence preprocessing was conducted by demultiplexing and trimming the adapter of the raw sequence using Cutadapt (v. 1.8.3) and sequence quality trimming using the SolexaQA (v.1.13) package. The cleaned reads were aligned to the Charleston gray watermelon reference genome (http://cucurbitgenomics.org, accessed on 14 August 2022) [22], and the alignment results were converted to BAM files using the Burrows–Wheeler alignment tool (BWA) program (v. 0.6.1-r104) [23]. SNP validation was conducted using the SEEDERS in-house script [24] based on the following conditions: minimum mapping quality for SNPs = 30, minimum mapping quality for gaps = 15, minimum read depth = 3, maximum read depth = 257, minimum indel score for nearby SNP filtering = 30, SNP within INT bp around a gap to be filtered = 30, and window size for filtering dense SNPs = 30. Raw SNPs were detected, and consensus sequences were extracted using the SAMtools program (v.0.1.16) and the SEEDERS in-house script.

### 4.5. SNP Discovery and Chromosome-Specific Distribution Analysis

A consolidated SNP matrix was created from the samples to perform a comparative analysis of the SNPs among samples. Raw SNP loci were obtained by comparing each sample with the reference genome, and a union list was constructed using these candidate loci. During this process, missing SNP loci were filled in from the consensus sequences of the samples to create the matrix. Subsequently, the SNP loci that were miscalled during sample comparison were filtered to create the final SNP matrix. Based on these loci, the SNPs were classified according to the following types: homozygous SNP: read rate ≥ 90%, heterozygous SNP: 40% ≤ read rate ≤ 60%, others: 20% ≤ read rate < 90%.

### 4.6. Identification of Introgression Region in Near Isogenic Lines (NILs) and Detection of Candidate Genes

To identify the introgression regions in each NIL sample, SNPs with the following traits were selected: those that exhibited nucleotide sequences identical to those of the donor parents (OTO9491 and NB5410) or those that were polymorphic to the recurrent parent (DAH). Chromosome-specific SNP distribution plots were created based on the identified SNPs, with a specific focus on homozygous SNPs. The genomic distribution of the filtered SNPs was visualized using the basic plotting function of the Circos program [25] in R v.2.1.5.2 (http://www.r-project.org). The candidate genes for orange-β flesh were selected from those involved in the carotenoids’ metabolic pathway and are located in the introgression regions and exhibit SNPs between the NILs.

### 4.7. Development of Candidate-Gene-Based Cleaved Amplified Polymorphic Sequences (CAPS) and Genotyping

Primer design and restriction enzyme search for CAPS were performed using Primer3 v.0.4.0 and NEB Cutter v.2.0. The derived-CAPS (dCAPS) marker was designed using the dCAPS Finder 2.0. The PCR mixture (10 µL) was prepared using 20 ng of genomic DNA, 1 µL 10 × PCR buffer, 0.2 mM dNTP, 0.5 µL 10 pmol forward primer, 0.5 µL 10 pmol reverse primer, 0.5 U Taq polymerase (Solgent, Daejeon, Republic of Korea), and distilled water. The PCR conditions were as follows: one cycle at 95 °C for 5 min, 35 cycles of denaturation at 94 °C for 30 s, annealing at the melting temperature of each primer for 30 s, extension at 72 °C for 1 min, and one cycle of final extension at 72 °C for 7 min. Restriction enzymes were added to the PCR amplicons according to the instructions of the manufacturer. Afterwards, the amplicons were electrophoresed using 3% agarose gel (Philekorea, Seoul, Republic of Korea) containing 3 µL of ethidium bromide per 100 mL at 200 V for 1 h. The gels were photographed under UV light using a Davinchi imaging system (Davinchi-K, Seoul, Republic of Korea). Genotyping of the *CRTISO* gene-based CAPS marker (CAPS-C^1976^) for orange flesh with a high prolycopene content (referred to as orange-p) was conducted according to the methods described by Jin et al. [13].

## 5. Conclusions

In summary, the substitution of Lys with Glu, a conserved amino acid alteration in *ClPSY1*, is associated with high β-carotene accumulation in orange-β-fleshed watermelons. The development of a CAPS marker using this SNP, in conjunction with *LCYB* and *CRTISO* gene-based markers, offers an effective means of distinguishing orange-β-fleshed watermelons from non-orange-β-fleshed watermelons. This study enhances our understanding of the genetic underpinnings governing β-carotene accumulation in watermelon flesh, and it also provides valuable molecular markers that enable marker-assisted selection of flesh color traits. However, further studies are required to identify whether a homologue of the orange gene is involved in the regulation of PSY in watermelons, and to uncover the functional role of the SNP of *ClPSY1* in enhancing β-carotene accumulation through the carotenoid biosynthesis pathway.

## Figures and Tables

**Figure 1 ijms-25-00210-f001:**
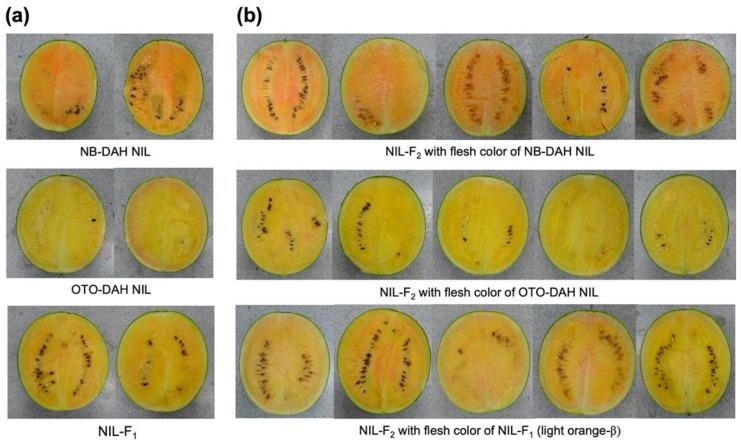
Flesh color of watermelon fruits used in this study. (**a**) Orange-β-fleshed (NB-DAH) near-isogenic line (NIL), canary-yellow-fleshed (OTO-DAH) NIL, and their F_1_ progeny (NIL-F_1_) with intermediate color (light orange-β). (**b**) F_2_ plants derived from self-pollination of the F_1_ progeny (NIL-F_1_). The fruits of five representative F_2_ plants for each flesh color category of NB-DAH-NIL (‘A’), OTO-DAH NIL (‘B’), and F_1_ (‘H’) are shown.

**Figure 2 ijms-25-00210-f002:**
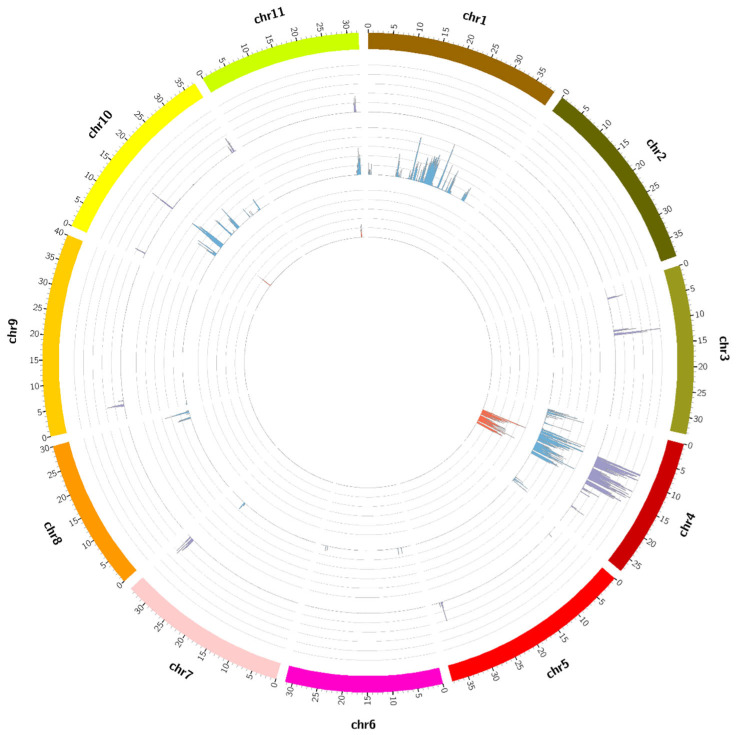
Circus plot representing physical distributions of homozygous single-nucleotide polymorphisms (SNPs) among the 11 watermelon chromosomes. SNP numbers were calculated based on their physical positions in a 1 Mb window size. First tract from inside: Distribution of 9520 SNPs (red) within the introgression regions shared by the NB-DAH NIL (orange-β-fleshed near-isogenic line) and OTO-DAH NIL (yellow-fleshed near-isogenic line). The *LCYB* gene located in this region is required for the expression of both canary yellow and orange flesh. Second tract: Distribution of 31,114 homozygous SNPs (blue) between DAH (red-fleshed recipient) and NB-DAH NIL. The genomic regions showing a high SNP frequency on chromosome 1 are the possible introgression regions from NB5410 (orange-*β*-fleshed donor for NB-DAH NIL); they, therefore, carry the gene(s) for orange-β-flesh. Among the 13 genes in the carotenoid metabolic pathway, only *PSY1* is located in this region. Third tract: Distribution of 16,755 homozygous SNPs between DAH (red-fleshed recipient) and OTO-DAH NIL. The outer track is the physical size of each chromosome represented in different colors.

**Figure 3 ijms-25-00210-f003:**
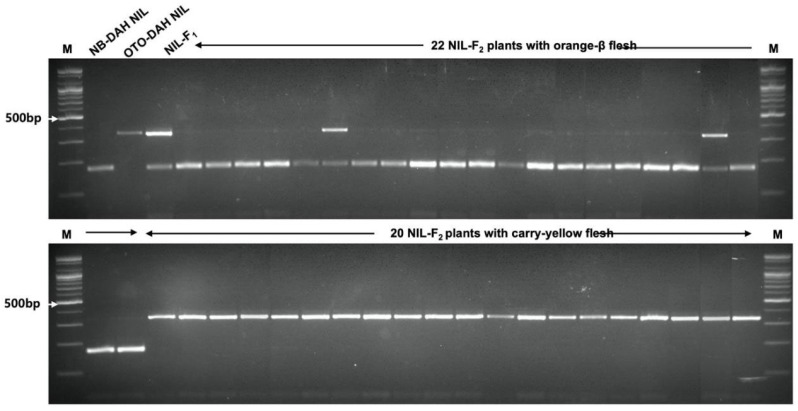
Agarose gel image of genotyping for 44 F_2_ plants (NIL-F_2_) derived from the cross between NB-DAH NIL and OTO-DAH NIL using the PSY1-A>G marker for watermelon fruit flesh color with a high β-carotene content (orange-β flesh).

**Figure 4 ijms-25-00210-f004:**
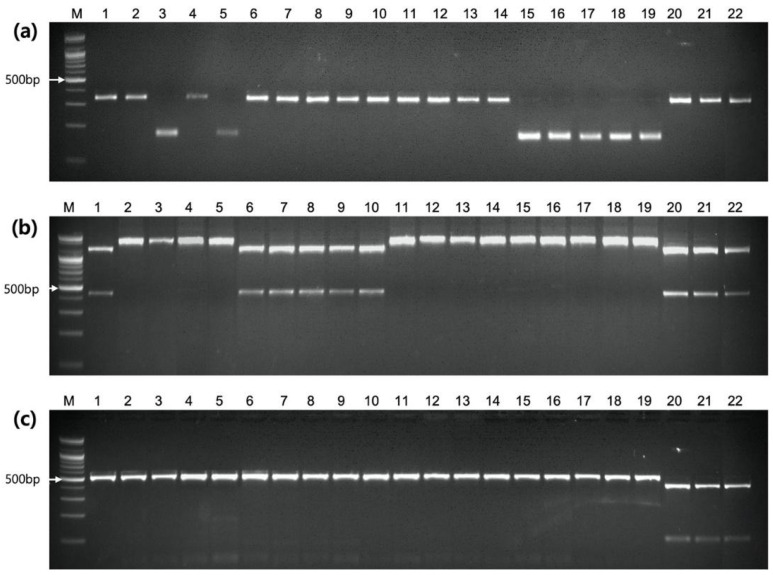
Agarose gel image of genotyping for 22 watermelon breeding lines and cultivars (Table 8) using three DNA markers for fruit flesh color, PSY1-A>G (**a**), LCYB-G>T (**b**), and CAPS-C^1976^ (**c**). M, 100 bp size maker; 1–22, entry number shown in Table 8.

**Figure 5 ijms-25-00210-f005:**
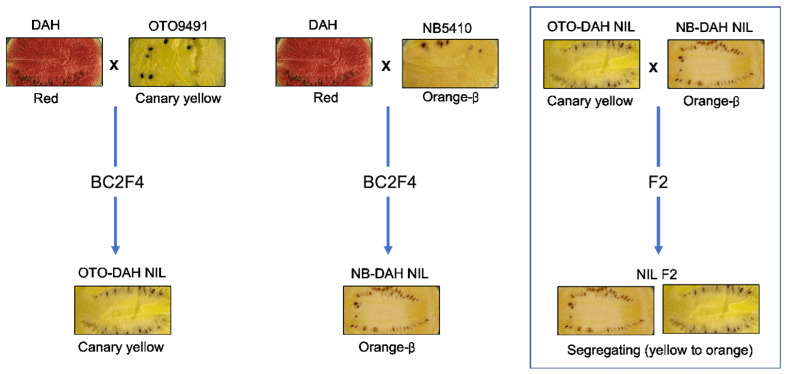
Fruit flesh colors of eight watermelon lines used in this study and the process of developing two near-isogenic lines (NILs) and an F_2_ population. Two rounds of backcrossing were performed, followed by three rounds of self-pollination to develop canary-yellow-fleshed (OTO-DAH NIL) and orange-β-fleshed (NB-DAH NIL) lines. Subsequently, OTO-DAH NIL and NB-DAH NIL were crossed to produce an F_1_ progeny; the F_1_ progeny was then self-pollinated to generate the F_2_ population.

**Table 1 ijms-25-00210-t001:** Flesh color and β-carotene contents in watermelon lines and populations.

Line and Population ^a^	Flesh Color ^b^	Generation	β-Carotene Content (mg/kg, Mean ± SD) ^c^
OTO9491	CY	Donor inbred	4.5 ± 1.2
NB5410	O-β	Donor inbred	128.4 ± 15.3
DAH	R	Recurrent inbred	10.3 ± 2.8
OTO-DAH NIL	CY	NIL (BC_2_F_4_)	15.5 ± 3.5
NB-DAH NIL	O-β	NIL (BC_2_F_4_)	135.4 ± 18.3
NIL-F_1_	Light O-β	F_1_	44.2 ± 5.8
NIL-F_2_	Segregated	F_2_	NA

^a^ NIL, near-isogenic line; ^b^ R, red; CY, canary yellow; O-β, orange with high β-carotene content; Light O-β, light orange-β; ^c^ mean ± SD, mean of three biological replicates ± standard deviation; NA, not available.

**Table 2 ijms-25-00210-t002:** Summarized results for whole-genome resequencing of the watermelon lines.

Line	No. of Reads	Total Length (bp)	Trimmed/Raw (%) ^a^	Genome Coverage ^b^
DAH	42,158,401	3,611,517,582	71.04	20.72
	42,158,401	3,640,042,936	71.60	
NB5410	23,469,008	1,984,647,849	67.91	11.27
	23,469,008	1,961,105,072	67.10	
OTO9491	22,586,239	1,878,291,403	59.59	10.3
	22,586,239	1,727,416,695	54.80	
NB-DAH NIL	20,030,848	1,699,640,123	67.76	9.72
	20,030,848	1,701,402,747	67.83	
OTO-D NIL	48,997,839	4,294,744,894	66.49	24.1
	48,997,839	4,141,524,303	64.11	

^a^ Trimmed/Raw: (total length of trimmed reads/total length of raw reads) × 100; ^b^ Approximate genome coverage: total read length of each sample divided by the estimated genome size (350 Mb).

**Table 3 ijms-25-00210-t003:** Number of single-nucleotide polymorphisms (SNPs) detected in the watermelon lines via read-mapping to the reference genome.

Sample	No. of Total SNPs	No. of Homozygous SNP ^a^	No. of Heterozygous SNP ^b^	No. of Other SNPs ^c^
DAH	245,860	209,526	10,643	25,691
NB5410	248,778	214,428	9292	25,058
OTO9491	246,534	216,062	8993	21,479
NB-DAH NIL	216,455	186,869	8673	20,913
OTO-DAH NIL	249,912	219,027	9631	21,254

^a^ When 90% of the mapped reads to the reference genome exhibited the same SNP type; ^b^ When 40–60% of the mapped reads to the reference genome showed the same SNP type, while the remaining reads exhibited an SNP type identical to the reference genome; ^c^ When classifying the reads as either homozygous or heterozygous was not possible.

**Table 4 ijms-25-00210-t004:** Number of single-nucleotide polymorphisms (SNPs) detected in the watermelon lines.

Line	No. of Polymorphic SNPs		No. of Other SNPs ^c^
	Homozygous SNP ^a^	Heterozygous SNP ^b^	
DAH vs. NB5410	147,343	5989	197,242
DAH vs. NB-DAH NIL	31,114	5347	256,284
DAH vs. OTO9491	142,498	5458	201,876
DAH vs. OTO-DAH NIL	16,755	3628	342,438
NB5410 vs. OTO9491	133,098	5584	196,707
NB-DAH NIL vs. OTO-DAH NIL	25,420	4887	262,148

^a^ When both samples contained homozygous nucleotide sequences for polymorphic SNPs; ^b^ When one sample contained a homozygous nucleotide sequence and the other contained a heterozygous nucleotide sequence for polymorphic SNPs; ^c^ Other SNPs include non-polymorphic SNPs (when the nucleotide sequences were identical between the compared samples), SNPs with insufficient depth (when at least one sample failed to meet the criterion of “depth ≥ 3” when comparing the nucleotide sequences of samples), unknown SNPs (when polymorphism could not be determined between samples due to missing data (missing genotype) in at least one sample during the comparison of nucleotide sequences), and ambiguous SNPs (when polymorphism could not be confirmed between samples due to the presence of “Etc.” SNP types in at least one sample during the comparison of nucleotide sequences).

**Table 5 ijms-25-00210-t005:** Genes encoding enzymes involved in the watermelon carotenoid synthesis pathway.

Gene ^a^	Encoded Enzyme	Gene ID ^b^	Gene Location ^b^	Introgression Region ^c^
*CHYB*	β-carotene hydroxylase	*ClCG01G002410*	Chr01: 2,378,461–2,381,614 (+)	-
*PSY*	Phytoene synthase	*ClCG01G008470*	Chr01: 10,039,958–10,042,905 (+)	NB-DAH NIL
		*ClCG02G023970*	Chr02: 38,304,958–38,307,096 (−)	-
		*ClCG07G010760*	Chr07: 26,633,212–26,636,449 (+)	-
*FPS*	Farnesyl diphosphate synthase	*ClCG01G011400*	Chr01: 18,713,844–18,718,625 (+)	-
		*ClCG09G021520*	Chr09: 38,485,275–38,489,544 (+)	-
*GGPR*	Geranylgeranyl diphosphate reductase	*ClCG02G024280*	Chr02: 38,557,070–38,558,437 (+)	-
*GGPS*	Geranylgeranyl diphosphate synthase	*ClCG02G003810*	Chr02: 3,851,659–3,854,438 (−)	-
*ZEP*	Zeaxanthin epoxidase	*ClCG02G012500*	Chr02: 25,999,338–26,006,398 (+)	-
		*ClCG02G024290*	Chr02: 38,562,422–38,564,200 (−)	-
*LCYB*	Lycopene β-cyclase	*ClCG04G004090*	Chr04: 15,694,446–15,696,571 (+)	NB-DAH NIL, OTO-DAH NIL
		*ClCG10G010860*	Chr10: 24,184,154–24,186,135 (+)	NB-DAH NIL
*ZDS*	ζ-carotene desaturase	*ClCG06G009140*	Chr06: 14,827,548–14,838,999 (−)	-
*PDS*	Phytoene desaturase	*ClCG07G015130*	Chr07: 31,562,036–31,578,907 (+)	-
*IPI*	Isopentenyl diphosphate isomerase	*ClCG09G003440*	Chr09: 2,959,189–3,006,838 (+)	-
*CRTISO*	Carotenoid isomerase	*ClCG10G017990*	Chr10: 32,958,299–32,963,939 (−)	-
		*ClCG10G007140*	Chr10: 10,280,425–10,285,209 (−)	-
*LCYE*	Lycopene ε-cyclase	*ClCG11G001670*	Chr11: 1,815,768–1,821,324 (+)	-

^a^ Genes listed in the order from upstream to downstream of the pathway; ^b^ Gene ID and location annotated based on the Charleston gray watermelon reference genome; (+), forward orientation; (−), reverse orientation; ^c^ Near-isogenic lines (NILs) carrying genes involved in the carotenoid metabolic pathway located within the genomic region introgressed from donor parental line.

**Table 6 ijms-25-00210-t006:** Single-nucleotide polymorphisms (SNPs) identified in carotenoid metabolism pathway genes located on the introgression region of near-isogenic lines (NILs).

Line	Flesh Color	Gene and SNP ^a^			
		*ClPSY1*	*ClLCYB*		
		Chr. 1	Chr. 4		
		10,040,402 bp	15,694,80 bp	15,695,470 bp	15,696,099 bp
		A>G (Non-syn)	A>G (Syn)	G>T (Non-syn)	G>C (Non-syn)
		Lys > Glu	-	Val > Phe	Lys > Asn
DAH	R	A	A	G	G
OTO9491	CY	A	G	T	C
NB5410	O-β	G	G	T	C
OTO-DAH NIL	CY	A	G	T	C
NB-DAH NIL	O-β	G	G	T	C

^a^ R, red; CY, canary yellow; O-β, orange with high-carotene content; Non-sys, non-synonymous SNP; Syn, synonymous SNP.

**Table 7 ijms-25-00210-t007:** List of DNA markers of fruit flesh color used in watermelon.

Marker	Type ^a^	Primer Sequence ^b^	Enzyme	PCR Band Size (bp) ^c^		
		(5ʹ–3ʹ)		CY	O-β	R
PSY1-A>G	CAPS	F: AGCAAGTATGGTGGCGAATC	*BcoD*I	338/43	175/163/43	338/43
		R: TAGCCTTTTGCCTCTCAGGA				
LCYB-G>T	CAPS	F: TGGAGAAAGCAAATTGAGCGAGCGATA	*BsaH*I	1594	1594	1180/414
		R: CCTGCTGTTCCACCAATTCCAACAACT				
LCYB-G>C	dCAPS	F: TTTTATTGAAGCTGGATCTTAA	*Afl*II	210	210	182/28
		R: TGCCGATCATGTTTACCAAA				

^a^ CAPS, cleaved amplified polymorphic sequence; dCAPS, derived leaved amplified polymorphic sequence; ^b^ F, forward primer; R, reverse primer; ^c^ R, red; CY, canary yellow; O-β, orange with high carotene content.

**Table 8 ijms-25-00210-t008:** Genotyping of 22 watermelon breeding lines and cultivars using three cleaved amplified polymorphic sequence markers associated with fruit flesh color.

EN	Line	Flesh Color ^a^	PSY1-A>G	LCYB-G>T	CAPS-C^1976^
1	DAH	R	AA	GG	TT
2	OTO9491	CY	AA	TT	TT
3	NB5410	O-β	GG	TT	TT
4	OTO-DAH NIL	CY	AA	TT	TT
5	NB-DAH NIL	O-β	GG	TT	TT
6	2962	R	AA	GG	TT
7	2966	R	AA	GG	TT
8	2987	R	AA	GG	TT
9	3005	R	AA	GG	TT
10	JB11-3	R	AA	GG	TT
11	Aldf	CY	AA	TT	TT
12	MgYF	CY	AA	TT	TT
13	YfDAH	CY	AA	TT	TT
14	YfSBA	CY	AA	TT	TT
15	DAHORF	O	GG	TT	TT
16	OfLTpd108	O	GG	TT	TT
17	MgOF	O	GG	TT	TT
18	OfSP43	O	GG	TT	TT
19	Summer Orange	O-β	GG	TT	TT
20	Golden Honey	O-p	AA	GG	CC
21	Orange Flesh Tender Sweet	O-p	AA	GG	CC
22	Tender Gold	O-p	AA	GG	CC

^a^ R, red; CY, canary yellow; O-β, orange with high β-carotene content; O-p, orange with high prolycopene content; O, orange with unknown carotenoid content.

## Data Availability

The WGRS data presented in the study are deposited in the National Center for Biotechnology Information (https://www.ncbi.nlm.nih.gov/) repository, accession number SRR26208948, SRR26208949, SRR26208950, SRR26208951, SRR26208952.

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
