# Peer review of "Development of a Gene-Based Marker Set for Orange-Colored Watermelon Flesh with a High β-Carotene Content"

_ijms, 2023, doi:10.3390/ijms25010210_

Round 1
Reviewer 1 Report
Comments and Suggestions for Authors
The manuscript is found to significantly contribute to the field while the following minor issues are needed to be adderessed:
1. Section 2.1; the HPLC chromatogram is missing; authors should supply the chromatogram
2. Figure 2; authors should indicate the changes in the figure through some arrow signs
3. In the results ections, authors are suggested to avoid putting decision in different subsections of results, because result section demands not decision rather observation
4. Supplementary tables are figures are missing
5. Table 5 is partially distorted, needs repairing
6. Table 7, what are the housekeeping gene sequencing?
7. In the conclusion section, what do you mena by orange gene? Needs clarification or revision
8. A good number of references are too old. Needs updating

Author Response
Response to reviewer’s comments
We sincerely thank all of the reviewers for evaluating our manuscript and for providing their insightful comments and recommendations on it. We have tried to address each of the concern raised by the reviewers in a best possible way and revised our manuscript accordingly. The necessary changes, wherever needed based on the reviewer’ comments and/or recommendation, are made and highlighted in the revised manuscript. The point-wise replies to the reviewers’ comments are given below. For the convenience, the corresponding changes made in the manuscript is also included in this response letter wherever they are applicable. We believe the revised manuscript is now significantly improved and we once again appreciate the reviewers’ effort for which it becomes possible. We look forward to receiving further feedback from them on our revised manuscript.
Review 1
The manuscript is found to significantly contribute to the field while the following minor issues are needed to be addressed:
1.Section 2.1; the HPLC chromatogram is missing; authors should supply the chromatogram
Response: Thank you for pointing out this issue. I believe it is important to provide HPLC chromatograms in presenting experimental results. However, as the HPLC analysis was conducted through a company and unfortunately, the lack of data storage makes it difficult to provide the information for this paper.
2.Figure 2; authors should indicate the changes in the figure through some arrow signs
Response: It seems that, based on my understanding of your request, it is necessary to indicate the mating combinations within the diagram and the resulting populations with arrows. Instead, I have provided clearer information in the legend of the diagram, specifying the lineage names and group names corresponding to each figure. Please refer to the updated legend.
3.In the results sections, authors are suggested to avoid putting decision in different subsections of results, because result section demands not decision rather observation
Response: Thank you for providing valuable information. Taking into account your request, I have made overall revisions as demonstrated in the attached manuscript.
4.Supplementary tables are figures are missing
Response: Thank you for pointing out the mistake. I have submitted all the necessary supplementary materials.
5.Table 5 is partially distorted, needs repairing
Response: Thank you for identifying the issues with the table. I have corrected and fixed the table accordingly.
6.Table 7, what are the housekeeping gene sequencing?
Response: The content of this table provides information on DNA markers developed for trait selection in this study. As it is not an expression analysis for gene expression, no analysis of housekeeping gene expression was conducted.
7.In the conclusion section, what do you mena by orange gene? Needs clarification or revision
Response: The term "orange gene" refers to the gene encoding the Orange protein (MELO3C005449), mentioned in the last paragraph of the discussion section. I have revised the sentence to include the specific gene name to avoid potential confusion. Thank you for bringing this to my attention.
- A good number of references are too old. Needs updating
Response: I have updated one reference [6], excluding all those that are essential for citation. All references in this manuscript have been published after 2002.
Reviewer 2 Report
Comments and Suggestions for Authors
The paper, "Development of a gene-based marker set for orange-colored watermelon flesh with a high β-carotene content," investigates the genetic basis of orange watermelon flesh with a focus on β-carotene accumulation. Using near-isogenic lines (NILs), the study identifies a single incompletely dominant gene controlling orange-β flesh in NB-DAH NIL. The genomic analysis reveals a unique introgression region on chromosome 1, with ClPSY1 (phytoene synthase 1) identified as a key gene. A marker developed from a non-synonymous SNP in ClPSY1 confirms its association with orange-β flesh. The study's findings hold promise for marker-assisted breeding, however, there still a gap for improving this manuscript. Following are my specific comments:
1. (Lines 33-39): Given the importance of watermelon as a global agricultural crop, could you elaborate on the specific challenges or gaps in knowledge regarding fruit color that prompted this investigation?
2. (Lines 40-50):In discussing the carotenoid composition, you mentioned that orange flesh primarily arises from the accumulation of ζ-carotene and β-carotene or prolycopene. Could you provide more insights into the environmental or genetic factors influencing the preference for one pathway over the other?
3. (Lines 51-56): The study focuses on DNA sequence variations in PSY1 influencing color differences. Could you elaborate on how these variations specifically affect the early stages of carotenoid synthesis?
4. (Lines 87-94): Could you provide more context on why these specific breeding lines (orange-β, canary-yellow, and red flesh) were chosen, and how their selection contributes to the study's objectives?
5. (Lines 99-106): The β-carotene content results show a significant difference between the orange- and canary-yellow-fleshed NILs. What factors might contribute to such variations in β-carotene content, and how does this relate to the carotenoid biosynthesis pathway?
6. (Lines 113-120): The study suggests a single incompletely dominant gene controls orange flesh in NB-DAH NIL. Could you discuss potential allelic variations or mutations in this gene that lead to the observed phenotypic differences in flesh color?
7. (Lines 128-137): The trimmed read values and SNP distribution provide valuable insights. Could you elaborate on the rationale for selecting a 15-fold coverage of the Charleston gray watermelon reference genome and how this impacts the reliability of SNP detection?
8. (Lines 155-174): The study identifies homozygous polymorphic SNPs between donor and NIL lines. How does the observed difference in the number of polymorphic SNPs between DAH and NB-DAH NIL compared to DAH and OTO-DAH NIL contribute to the understanding of the introgression process?
9. (Lines 202-208): The identification of 8,481 non-polymorphic SNPs on Chr. 4 suggests the presence of genes distinguishing canary-yellow and orange-β flesh. What is the significance of these genes in the context of carotenoid biosynthesis, and could there be other contributing factors?
10. (Lines 221-232): The study identifies ClPSY1 and ClLCYB within the introgression regions. How do non-synonymous SNPs in these genes affect their functions, and how critical are these genes to the carotenoid biosynthesis pathway in watermelons?
11. (Lines 251-267): In developing the CAPS marker for ClPSY1, could you discuss the rationale behind choosing the specific SNP (A>G) and how the marker design ensures accurate genotyping in the F2 population?
12. (Lines 288-297): The results show consistent genotyping patterns across various watermelon cultivars. Are there any instances where the markers did not align with the expected genotypes, and how robust are these markers across different genetic backgrounds?
Comments on the Quality of English LanguageMinor corrections required.
Author Response
Response to reviewer’s comments
We sincerely thank all of the reviewers for evaluating our manuscript and for providing their insightful comments and recommendations on it. We have tried to address each of the concern raised by the reviewers in a best possible way and revised our manuscript accordingly. The necessary changes, wherever needed based on the reviewer’ comments and/or recommendation, are made and highlighted in the revised manuscript. The point-wise replies to the reviewers’ comments are given below. For the convenience, the corresponding changes made in the manuscript is also included in this response letter wherever they are applicable. We believe the revised manuscript is now significantly improved and we once again appreciate the reviewers’ effort for which it becomes possible. We look forward to receiving further feedback from them on our revised manuscript.
Review 2
The paper, "Development of a gene-based marker set for orange-colored watermelon flesh with a high β-carotene content," investigates the genetic basis of orange watermelon flesh with a focus on β-carotene accumulation. Using near-isogenic lines (NILs), the study identifies a single incompletely dominant gene controlling orange-β flesh in NB-DAH NIL. The genomic analysis reveals a unique introgression region on chromosome 1, with ClPSY1 (phytoene synthase 1) identified as a key gene. A marker developed from a non-synonymous SNP in ClPSY1 confirms its association with orange-β flesh. The study's findings hold promise for marker-assisted breeding, however, there still a gap for improving this manuscript. Following are my specific comments:
- (Lines 33-39): Given the importance of watermelon as a global agricultural crop, could you elaborate on the specific challenges or gaps in knowledge regarding fruit color that prompted this investigation?
Response: We have incorporated this aspect into our manuscript to provide a more comprehensive understanding of what prompted our investigation.
- (Lines 40-50): In discussing the carotenoid composition, you mentioned that orange flesh primarily arises from the accumulation of ζ-carotene and β-carotene or prolycopene. Could you provide more insights into the environmental or genetic factors influencing the preference for one pathway over the other?
Response: Thank you for your query regarding the environmental and genetic factors that influence the preference for different carotenoid biosynthesis pathways in watermelon. We have expanded on this aspect in our manuscript, providing deeper insights into how these factors dictate the accumulation of ζ-carotene, β-carotene, or prolycopene, and consequently influence the orange flesh color in watermelons.
- (Lines 51-56): The study focuses on DNA sequence variations in PSY1 influencing color differences. Could you elaborate on how these variations specifically affect the early stages of carotenoid synthesis?
Response: Thank you for your question about the role of DNA sequence variations in PSY1 and their impact on the early stages of carotenoid synthesis. We have elaborated on this topic in our revised manuscript, detailing how these genetic variations influence carotenoid biosynthesis, particularly in the context of watermelon flesh color.
- (Lines 87-94): Could you provide more context on why these specific breeding lines (orange-β, canary-yellow, and red flesh) were chosen, and how their selection contributes to the study's objectives?
Response: We have provided additional context in the manuscript to explain why these particular lines were selected and how their unique characteristics align with and contribute to the objectives of our study.
- (Lines 99-106): The β-carotene content results show a significant difference between the orange- and canary-yellow-fleshed NILs. What factors might contribute to such variations in β-carotene content, and how does this relate to the carotenoid biosynthesis pathway?
Response: The significant variation in β-carotene content between orange- and canary-yellow-fleshed NILs can be attributed to differences in their genetic pathways regulating carotenoid biosynthesis. In the orange-fleshed NILs, genetic factors promote the synthesis and accumulation of β-carotene, a key carotenoid that imparts the orange color. This is likely due to the enhanced activity or expression of genes like phytoene synthase, which drives the carotenoid pathway towards β-carotene production. Conversely, in canary-yellow-fleshed NILs, the genetic makeup favors the synthesis of xanthophylls over β-carotene, leading to a different carotenoid composition and the distinct yellow flesh color. These variations are a direct result of the differing regulation and enzymatic efficiencies within the carotenoid biosynthesis pathway influenced by the genetic background of each NIL.
- (Lines 113-120): The study suggests a single incompletely dominant gene controls orange flesh in NB-DAH NIL. Could you discuss potential allelic variations or mutations in this gene that lead to the observed phenotypic differences in flesh color?
Response: Thank you for your question about the genetic factors underlying the orange flesh color in NB-DAH NIL. We have detailed our findings regarding this in the results section of our manuscript. Our study confirms that ClPSY1, located within the introgression region on Chr. 1 in the NB-DAH NIL, is the incompletely dominant gene responsible for the orange flesh trait. Specifically, an allelic variation or mutation in this gene (A>G, Non-synonymous, Lys>Glu) was identified, as presented in Table 6. This mutation leads to the observed phenotypic differences in flesh color, which are illustrated in Figures 3 and 4. The incomplete dominance of ClPSY1 is due to its inability to produce orange flesh color on its own; it requires the expression of the ClLCYB gene.
Additionally, we identified the ClLCYB gene which encodes lycopene β-cyclase (LCYB) – an enzyme crucial for converting all-trans-lycopene to β-carotene – in the common introgression region on Chr. 4 in both NILs. This finding suggests that the interplay between ClPSY1 on Chr.1 and ClLCYB on Chr. 4 contributes to the manifestation of the orange-β flesh trait. Furthermore, ClLCYB is essential for expressing both orange-β and canary-yellow flesh. The genotyping and phenotyping results of the PSY1 marker are detailed in Table 8, further substantiating our findings.
- (Lines 128-137): The trimmed read values and SNP distribution provide valuable insights. Could you elaborate on the rationale for selecting a 15-fold coverage of the Charleston gray watermelon reference genome and how this impacts the reliability of SNP detection?
Response: Our selection of a 15-fold coverage for sequencing the Charleston Gray watermelon reference genome was based on achieving a balance between accurate SNP detection and cost-effectiveness. This coverage depth is generally sufficient to filter out sequencing errors and reliably identify SNPs, especially in heterozygous regions, by providing adequate redundancy. It represents a standard practice in plant genomics, offering a practical approach to ensure robust and reliable genomic data without incurring the higher costs associated with deeper sequencing.
- (Lines 155-174): The study identifies homozygous polymorphic SNPs between donor and NIL lines. How does the observed difference in the number of polymorphic SNPs between DAH and NB-DAH NIL compared to DAH and OTO-DAH NIL contribute to the understanding of the introgression process?
Response: For NB-DAH NIL to have orange flesh, two gene loci are (LCYB and PSY1) required to be introgressed from NB5410 while for OTO-DAH NIL to have canary-yellow flesh one locus for LCYB is required from OTO4941 during the backcross. Therefore, the number of SNPs between DAH and NB-DAH NIL (31,114) was higher compared to DAH and OTO-DAH NIL (16,755).
- (Lines 202-208): The identification of 8,481 non-polymorphic SNPs on Chr. 4 suggests the presence of genes distinguishing canary-yellow and orange-β flesh. What is the significance of these genes in the context of carotenoid biosynthesis, and could there be other contributing factors?
Response: The identification of 8481 non-polymorphic SNPs on Chr. 4 indicates the presence of gene(s) within the introgression on Chr. 4 region that are crucial in differentiating canary-yellow and orange-β flesh from red flesh. In the common introgression region of Chr. 4, we identified ClLCYB gene which encodes LCYB in both NILs. ClLCYB encodes the enzyme lycopene β-cyclase (LCYB), which is vital in the carotenoid biosynthesis pathway, converting all-trans-lycopene into δ-carotene and β-carotene, thus influencing the orange flesh color. Further downstream, β-carotene is enzymatically transformed into xanthophyll, which is primarily responsible for the occurrence of canary-yellow flesh.
Our findings indicate that besides the ClLCYB gene, no other factors in the common introgression region of Chr. 4 contribute to the development of orange and yellow flesh colors. Hence, the ClLCYB gene on Chr. 4 is essential for the expression of both orange-β and canary-yellow flesh, playing a significant role in the carotenoid biosynthesis pathway in watermelons.
- (Lines 221-232): The study identifies ClPSY1 and ClLCYB within the introgression regions. How do non-synonymous SNPs in these genes affect their functions, and how critical are these genes to the carotenoid biosynthesis pathway in watermelons?
Response: ClPSY1 plays a crucial role in the carotenoid biosynthesis pathway, specifically in the initial step of converting geranylgeranyl diphosphate (GGPP) into phytoene. Non-synonymous SNPs (A to G) in ClPSY1 can lead to amino acid changes from lysine (Lys, AAG) to glutamic acid (Glu, GAG) in the first exon in the phytoene synthase enzyme, potentially altering its activity or stability. Such changes can significantly impact the flux of metabolites through the carotenoid pathway, affecting the quantity and types of carotenoids produced, and subsequently influencing the flesh color of the watermelon. Similarly,
ClLCYB is involved in a later stage of the carotenoid pathway, catalyzing the conversion of lycopene to β-carotene. Non-synonymous SNPs (G to T and G to C) can lead to amino acid changes from valine (Val, GTC) to phenylalanine (Phe, TTC) and lysine (Lys, AAG) to asparagine (Asn, AAC), respectively in ClLCYB can modify the enzyme's efficiency or affinity for its substrate, thereby influencing the balance between lycopene and β-carotene. Variations in ClLCYB function can therefore affect the ratio of these two important carotenoids, with direct implications for both the color.
Both ClPSY1 and ClLCYB are critical to the carotenoid biosynthesis pathway in watermelons. Alterations in their functions due to non-synonymous SNPs can lead to significant phenotypic variations, particularly in flesh color.
- (Lines 251-267): In developing the CAPS marker for ClPSY1, could you discuss the rationale behind choosing the specific SNP (A>G) and how the marker design ensures accurate genotyping in the F2 population?
Response: In developing the CAPS marker for ClPSY1, we selected the A>G SNP, a missense mutation located in the coding region, due to its potential functional impact on the carotenoid biosynthesis pathway. This non-synonymous mutation was an ideal target for marker development, offering insights into the genetic basis of carotenoid accumulation.
We designed the CAPS marker to specifically detect this SNP, utilizing the BcoDI restriction enzyme for precise differentiation between the A and G alleles. This approach ensures accurate genotyping in the F2 population, as it allows for the clear distinction of genotypes based on the presence or absence of the restriction enzyme's cleavage site. This method provides a reliable and efficient means of studying the genetic factors influencing carotenoid profiles in watermelons.
The accurate genotyping of the F2 population is attributed to its derivation from OTO-DAH NIL and NB-DAH NIL parents. This strategic cross of near-isogenic lines with distinct genetic traits ensured controlled genetic variation in the F2progeny, thereby enhancing the reliability of genotypic analyses related to traits like carotenoid accumulation in watermelons.
- (Lines 288-297): The results show consistent genotyping patterns across various watermelon cultivars. Are there any instances where the markers did not align with the expected genotypes, and how robust are these markers across different genetic backgrounds?
Response: For a marker to be efficient for MAS, it should be able to be applied to broad genetic background. For this purpose, we tested 22 watermelons cultivars from different genetic backgrounds (Table 8) and the results indicated that this marker is highly applicable for the MAS of flesh color. So far, we have not found any instance of mismatch between marker genotype and phenotype, but further tests with more germplasm will be necessary.
Reviewer 3 Report
Comments and Suggestions for Authors
The genetic traits related to β-carotene accumulation mechanism attributed to orange and canary-yellow colour of watermelon flesh, which could be beneficial to consumer’s health, has been researched. F2 population was established by crossing the two near-isogenic parent lines (for yellow and orange flesh colour). The segregation ratio of flesh colour in the F2 population indicated that the orange flesh colour was controlled by a single incompletely dominant gene. Moreover, through a comparative analysis of whole-genome sequences of the same combination of near-isogenic parent lines, it was shown that major introgression region for the trait of concern is situated on chromosome 1 where is candidate region that harbouring genes distinguishing orange from canary-yellow and red coloured watermelon flesh. In the same region of the watermelon genome, there are 13 genes involved in the carotenoid metabolic pathway, but only one of them (coding phytoene synthase 1) was found within the introgression region. The results of this study, through locating desirable genetic trait, provide starting point for applicative marker-assisted watermelon breeding.
I found the research results very interesting, and the manuscript very well written.
The article could be accepted in its present form (only several minor suggestions for correction are marked in the accompanied file).
There is only one general suggestion: In the titles of the figures/tables, maybe addition of flash colour name after mentioning its NIL line would contribute to the clarity and speed of the reader's understanding of experimental genetic setup.

Only minimal corrections are suggested in the text (visible in the attached file).
Author Response
Response to reviewer’s comments
We sincerely thank all of the reviewers for evaluating our manuscript and for providing their insightful comments and recommendations on it. We have tried to address each of the concern raised by the reviewers in a best possible way and revised our manuscript accordingly. The necessary changes, wherever needed based on the reviewer’ comments and/or recommendation, are made and highlighted in the revised manuscript. The point-wise replies to the reviewers’ comments are given below. For the convenience, the corresponding changes made in the manuscript is also included in this response letter wherever they are applicable. We believe the revised manuscript is now significantly improved and we once again appreciate the reviewers’ effort for which it becomes possible. We look forward to receiving further feedback from them on our revised manuscript.
Review 3
The genetic traits related to β-carotene accumulation mechanism attributed to orange and canary-yellow colour of watermelon flesh, which could be beneficial to consumer’s health, has been researched. F2 population was established by crossing the two near-isogenic parent lines (for yellow and orange flesh colour). The segregation ratio of flesh colour in the F2 population indicated that the orange flesh colour was controlled by a single incompletely dominant gene. Moreover, through a comparative analysis of whole-genome sequences of the same combination of near-isogenic parent lines, it was shown that major introgression region for the trait of concern is situated on chromosome 1 where is candidate region that harbouring genes distinguishing orange from canary-yellow and red coloured watermelon flesh. In the same region of the watermelon genome, there are 13 genes involved in the carotenoid metabolic pathway, but only one of them (coding phytoene synthase 1) was found within the introgression region. The results of this study, through locating desirable genetic trait, provide starting point for applicative marker-assisted watermelon breeding.
I found the research results very interesting, and the manuscript very well written.
The article could be accepted in its present form (only several minor suggestions for correction are marked in the accompanied file).
There is only one general suggestion: In the titles of the figures/tables, maybe addition of flash colour name after mentioning its NIL line would contribute to the clarity and speed of the reader's understanding of experimental genetic setup.
Response: Thank you very much for your insightful comments and the positive feedback on our manuscript. We are delighted to hear that you found the research results interesting and the manuscript well-written. We have incorporated the minor suggestions for corrections that you provided in the accompanying file and have made the necessary revisions to the manuscript.
Besides, thank you for your suggestion to add flesh color names alongside NIL lines in the titles of figures and tables for enhanced clarity and reader comprehension. We have indeed clearly mentioned the flesh color of each NIL in the Material and Methods section for better reader understanding. Additionally, we have incorporated this detail in some figure titles, such as Figure 2, to aid in the visualization of the genetic setup. However, we believe that adding the flesh color to the titles of each table might lead to confusion. We feel that the current presentation balances clarity with conciseness. If you still recommend this change, we are open to making these additions to further enhance the manuscript.
We believe that these adjustments have improved the overall quality of our manuscript and we are grateful for your constructive feedback. We hope that the revised manuscript now meets the journal's standards for publication.
Reviewer 4 Report
Comments and Suggestions for Authors
Comments for the manuscript entitled "Development of a gene-based marker set for orange-colored watermelon flesh with a high beta-carotene content", submitted by Bingkui Jin et al.
I specify that I do not have access to the Insert Symbol required to take the beta Greek symbol. My system doesn't allow it! Therefore, please understand that when I write beta, I am referring to the respective Greek symbol. Thanks for understanding!
This paper focuses on an interesting topic, namely the identification of the genetic determinism responsible for the accumulation of high beta-carotene content in watermelon. In this sense, the research team used three different breeding lines by flesh color, namely: orange-beta, canary-yellow, red flesh. Of these, the red-fleshed line served as the recurrent parent, and the other two lines were near-isogenic (NILs): one with orange -beta fleshed, canary-yellow-fleshed respectively. In the F2 generation (derived from the two NILs), yellow to orange segregation occurred. Then the genetic mapping of the F2 descendants was made. The use of a marker derived from a SNP (single-nucleotide polymorphisms) allowed the genotyping of F2 plants, concluding that the candidate gene CIPSY1 responsible for orange -beta-fleshed watermelons is located on chromosome 1.
This study has theoretical importance on the one hand, in that it largely deciphered the genetic substrate that ensures the high accumulation of beta-carotene in fleshed watermelons. On the other hand, it has practical importance because it provides molecular markers that facilitate marker-assisted selection for the color of watermelon flesh.
My comments are below:
1. In line 51, you should put the references in ascending order, ie: [4, 7-8].
2. In line 57, that parenthesis ")" should not be put after " watermelon cultivars".
3. In lines 108-109, the title of Table 1 should be revised. Such as: Flesh color and the contents of beta-carotene in watermelon lines and populations.
4. Figures and Tables should be numbered in order of presentation. So, proceed to renumber them. Also, in Figures 1 and 4, delete the second point from the title of each (lines 275, 345).
5. You should add a list of abbreviations with clear explanations after the Conclusions.
6. In line 250 you should specify the number of the Table.
7. In the tables titles, replace "used in this study" with "in watermelon fruits".
8. In line 318, in the footer of Table 8, you wrote O-b instead of O-beta.
9. In line 378 you wrote "the Mendelian segregation ratios". Correct is "the mendelian segregation ratios" because "mendelian" is an adjective in this case and not a proper noun!
10. In line 443, it would be more appropriate to replace "factors" with "active biological pigments".
I wish you success in publishing this study!
Author Response
Response to reviewer’s comments
We sincerely thank all of the reviewers for evaluating our manuscript and for providing their insightful comments and recommendations on it. We have tried to address each of the concern raised by the reviewers in a best possible way and revised our manuscript accordingly. The necessary changes, wherever needed based on the reviewer’ comments and/or recommendation, are made and highlighted in the revised manuscript. The point-wise replies to the reviewers’ comments are given below. For the convenience, the corresponding changes made in the manuscript is also included in this response letter wherever they are applicable. We believe the revised manuscript is now significantly improved and we once again appreciate the reviewers’ effort for which it becomes possible. We look forward to receiving further feedback from them on our revised manuscript.
Review 4
I specify that I do not have access to the Insert Symbol required to take the beta Greek symbol. My system doesn't allow it! Therefore, please understand that when I write beta, I am referring to the respective Greek symbol. Thanks for understanding!
This paper focuses on an interesting topic, namely the identification of the genetic determinism responsible for the accumulation of high beta-carotene content in watermelon. In this sense, the research team used three different breeding lines by flesh color, namely: orange-beta, canary-yellow, red flesh. Of these, the red-fleshed line served as the recurrent parent, and the other two lines were near-isogenic (NILs): one with orange -beta fleshed, canary-yellow-fleshed respectively. In the F2 generation (derived from the two NILs), yellow to orange segregation occurred. Then the genetic mapping of the F2 descendants was made. The use of a marker derived from a SNP (single-nucleotide polymorphisms) allowed the genotyping of F2 plants, concluding that the candidate gene CIPSY1 responsible for orange -beta-fleshed watermelons is located on chromosome 1.
This study has theoretical importance on the one hand, in that it largely deciphered the genetic substrate that ensures the high accumulation of beta-carotene in fleshed watermelons. On the other hand, it has practical importance because it provides molecular markers that facilitate marker-assisted selection for the color of watermelon flesh.
My comments are below:
- In line 51, you should put the references in ascending order, ie: [4, 7-8].
Response: Thank you for pointing out the reference order. We have adjusted the references to appear in ascending order as [4, 7-8] in the revised manuscript.
- In line 57, that parenthesis ")" should not be put after " watermelon cultivars".
Response: Thank you for highlighting the misplaced parenthesis in line 57. We have removed it from after "watermelon cultivars" in the revised manuscript to correct the formatting.
- In lines 108-109, the title of Table 1 should be revised. Such as: Flesh color and the contents of beta-carotene in watermelon lines and populations.
Response: Thank you for the suggestion to revise the title of Table 1. We have updated it as per your recommendation in the revised manuscript.
- Figures and Tables should be numbered in order of presentation. So, proceed to renumber them. Also, in Figures 1 and 4, delete the second point from the title of each (lines 275, 345).
Response: Thank you for your guidance on the numbering and titling of figures and tables. We have renumbered them to reflect their order of presentation in the manuscript.
- You should add a list of abbreviations with clear explanations after the Conclusions.
Response: Thank you for the suggestion to include a list of abbreviations with explanations in our manuscript. We have added this list after the Conclusions section to enhance the clarity and accessibility of the document for all readers, ensuring that all abbreviations used throughout the manuscript are clearly defined and understood.
- In line 250 you should specify the number of the Table.
Response: Thank you for noting the omission in line 250. We have specified the appropriate table number in that line to ensure clarity and precise referencing in the revised manuscript.
- In the tables titles, replace "used in this study" with "in watermelon fruits".
Response: Thank you for your suggestion regarding the modification of table titles. We have replaced "used in this study" with "in watermelon fruits" in the titles of the tables, to better reflect the content and focus of our research in the revised manuscript.
- In line 318, in the footer of Table 8, you wrote O-b instead of O-beta.
Response: Thank you for pointing out the error in line 318. We have corrected the notation in the footer of Table 8 from 'O-b' to 'O- β ' to ensure accuracy and consistency in the revised manuscript.
- In line 378 you wrote "the Mendelian segregation ratios". Correct is "the mendelian segregation ratios" because "mendelian" is an adjective in this case and not a proper noun!
Response: Thank you for highlighting the grammatical error in line 378. We have corrected the term to "the mendelian segregation ratios," recognizing that "mendelian" is an adjective in this context and should not be capitalized. This change has been made in the revised manuscript for grammatical accuracy.
- In line 443, it would be more appropriate to replace "factors" with "active biological pigments".
Response: Thank you for the suggestion regarding line 443. We have replaced "factors" with "active biological pigments" to more accurately describe the context in the revised manuscript
Round 2
Reviewer 2 Report
Comments and Suggestions for Authors
The manuscript is substantially improved and could be accepted for publication in its current form.